# Unified Human-Scene Interaction via Prompted Chain-of-Contacts

**Zeqi Xiao**[1,2]**, Tai Wang**[1]**, Jingbo Wang**[1]**, Jinkun Cao**[1,3]**, Wenwei Zhang**[1]**, Bo Dai**[1]**,
Dahua Lin**[1]**, Jiangmiao Pang**[1✉]
[1]Shanghai AI Laboratory, [2]S-Lab, NTU, [3]CMU

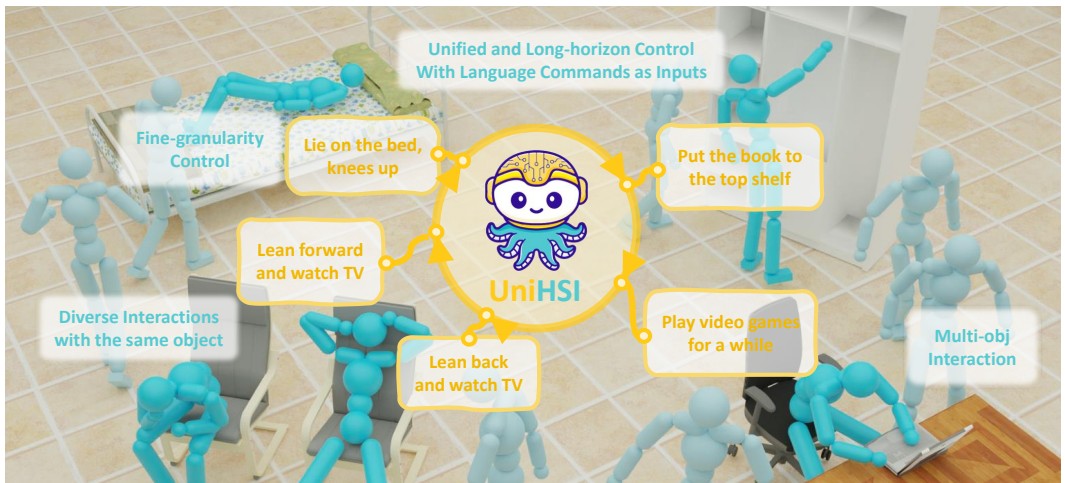

Figure 1: UniHSI facilitates unified and long-horizon control in response to natural language commands, offering notable features such as diverse interactions with a singular object, multi-object interactions, and fine-granularity control.

## Abstract

Human-Scene Interaction (HSI) is a vital component of fields like embodied AI and virtual reality. Despite advancements in motion quality and physical plausibility, two pivotal factors, versatile interaction control and user-friendly interfaces, require further exploration for the practical application of HSI. This paper presents a unified HSI framework, named *UniHSI*, that supports unified control of diverse interactions through language commands. The framework defines interaction as "Chain of Contacts (CoC)", representing steps involving human joint-object part pairs. This concept is inspired by the strong correlation between interaction types and corresponding contact regions. Based on the definition, UniHSI constitutes a *Large Language Model (LLM) Planner* to translate language prompts into task plans in the form of CoC, and a *Unified Controller* that turns CoC into uniform task execution. To support training and evaluation, we collect a new dataset named *ScenePlan* that encompasses thousands of task plans generated by LLMs based on diverse scenarios. Comprehensive experiments demonstrate the effectiveness of our framework in versatile task execution and generalizability to real scanned scenes.

## 1 Introduction

Human-Scene Interaction (HSI) constitutes a crucial element in various applications, including embodied AI and virtual reality. Despite the great efforts in this domain to promote motion quality (Holden et al., 2017; Starke et al., 2019; 2020; Hassan et al., 2021b; Zhao et al., 2022; Hassan et al.,

---

✉ Corresponding Author. Project page at this URL.

2021a; Wang et al., 2022a) and physical plausibility (Holden et al., 2017; Starke et al., 2019; 2020; Hassan et al., 2021b; Zhao et al., 2022; Hassan et al., 2021a; Wang et al., 2022a), two key factors, versatile interaction control and the development of a user-friendly interface, are yet to be explored before HSI can be put into practical usage.

This paper aims to provide an HSI system that supports versatile interaction control through language commands, one of the most uniform and accessible interfaces for users. Such a system requires: 1) Aligning language commands with precise interaction execution, 2) Unifying diverse interactions within a single model to ensure scalability. To achieve this, the initial effort involves the uniform definition of different interactions. We propose that interaction itself contains a strong prior in the form of human-object contact regions. For example, in the case of "lie down on the bed", it can be interpreted as "first the pelvis contacting the mattress of the bed, then the head contacting the pillow". To this end, we formulate interaction as ordered sequences of human joint-object part contact pairs, which we refer to as *Chain of Contacts (CoC)*. Unlike previous contact-driven methods, which are limited to supporting specific interactions through manual design, our interaction definition is generalizable to versatile interactions and capable of modeling multi-round transitions. The recent advancements in Large Language Models have made it possible to translate language commands into CoC. The structured formulation then can be uniformly processed for the downstream controller to execute.

Following the above formulation, we propose **UniHSI**, the first **Uni**fied physical **HSI** framework with language commands as inputs. UniHSI consists of a high-level **LLM Planner** to translate language inputs into the task plans in the form of CoC and a low-level **Unified Controller** for executing these plans. Combining language commands and background information such as body joint names and object part layout, we harness prompt engineering techniques to instruct LLMs to plan interaction step by step. We design the TaskParser to support the unified execution. It serves as the core of the Unified Controller. Following CoC, the TaskParser collects information including joint poses and object point clouds from the physical environment, then formulates them into uniform task observations and task objectives.

As illustrated in Fig. 1, the Unified Controller models whole-body joints and arbitrary parts of objects in the scenarios to enable fine-granularity control and multi-object interaction. With different language commands, we can generate diverse interactions with the same object. Unlike previous methods that only model a limited horizon of interactions, like "sitting down", we design the TaskParser to evaluate the completion of the current steps and sequentially fetch the next step, resulting in multi-round and long-horizon transition control. The Unified control leverages the adversarial motion prior framework (Peng et al., 2021) that uses a motion discriminator for realistic motion synthesis and a physical simulation (Makoviychuk et al., 2021) to ensure physical plausibility.

Another impressive feature of our framework is the training is interaction annotation-free. Previous methods typically require datasets that capture both target objects and the corresponding motion sequences, which demand numerous laboring. In contrast, we leverage the interaction knowledge of LLMs to generate interaction plans. It significantly reduces the annotation requirements and makes versatile interaction training feasible. To this end, we create a novel dataset named **ScenePlan**. It encompasses thousands of interaction plans based on scenarios constructed from PartNet (Mo et al., 2019) and ScanNet (Dai et al., 2017) datasets. We conduct comprehensive experiments on ScenePlan. The results illustrate the effectiveness of the model in versatile interaction control and good generalizability on real scanned scenarios.

## 2 RELATED WORKS

**Kinematics-based Human-Scene Interaction.** How to synthesize realistic human behavior is a long-standing topic. Most existing methods focus on promoting the quality and diversity of humanoid movements (Barsoum et al., 2018; Harvey et al., 2020; Pavllo et al., 2018; Yan et al., 2019; Zhang et al., 2022a; Tevet et al., 2022b; Zhang et al., 2023b) but do not consider scene influence. Recently, there has been a growing interest in synthesizing motion with human-scene interactions, driven by its applications in various applications like embodied AI and virtual reality. Many previous methods (Holden et al., 2017; Starke et al., 2019; 2020; Hassan et al., 2021b; Zhao et al., 2022; Hassan et al., 2021a; Wang et al., 2022a; Zhang et al., 2022b; Wang et al., 2022b) use data-driven kinematic models to generate static or dynamic interactions. These methods are typically inferior in

Table 1: Comparative Analysis of Key Features between UniHSI and Preceding Methods.

| Methods | Unified Interaction | Language Input | Long-horizon Transition | Interaction Annotation-free | Control Joints | Multi-object Interactions |
|---|---|---|---|---|---|---|
| NSM Starke et al. (2019) | | | ✓ | | 3 (pelvis, hands) | ✓ |
| SAMP Hassan et al. (2021a) | | | | | 1 (pelvis) | |
| COUCH Zhang et al. (2022b) | | | | | 3 (pelvis, hands) | ✓ |
| HUMANISE Wang et al. (2022b) | ✓ | ✓ | | | - | |
| ScenDiffuser Huang et al. (2023) | ✓ | ✓ | | | - | |
| PADL Juravsky et al. (2022) | | ✓ | ✓ | ✓ | - | |
| InterPhys Hassan et al. (2023) | | | | | 4 (pelvis, head, hands) | |
| Ours | ✓ | ✓ | ✓ | ✓ | 15 (whole-body) | ✓ |

physical plausibility and prone to synthesizing motions with artifacts, such as penetration, floating, and sliding. The need for additional post-processing to mitigate these artifacts hinders the real-time applicability of these frameworks.

**Physics-based Human-Scene Interaction.** Recent advances in physics-based methods (e.g., (Peng et al., 2021; 2022; Hassan et al., 2023; Juravsky et al., 2022; Pan et al., 2023) hold promise for ensuring physical realism through physics-aware simulators. However, they have limitations: 1) They typically require separate policy networks for each task, limiting their ability to learn versatile interactions within a unified controller. 2) These methods often focus on basic action-based control, neglecting finer-grained interaction details. 3) They heavily rely on annotated motion sequences for human-scene interactions, which can be challenging to obtain. In contrast, our UniHSI re-designs human-scene interactions into a uniform representation, driven by world knowledge from our high-level LLM Planner. This allows us to train a unified controller with versatile interaction skills without the need for annotated motion sequences. Key feature comparisons are in Tab. 1.

**Languages in Human Motion Control.** Incorporating language understanding into human motion control has become a recent research focus. Existing methods primarily focus on scene-agnostic motion synthesis (Zhang et al., 2022a; Chen et al., 2023; Tevet et al., 2022a;b; Zhang et al., 2023a;b; Jiang et al., 2023) (Athanasiou et al., 2023). Generating human-scene interactions using language commands poses additional challenges because the output movements must align with the commands and be coherent with the environment. Zhao et al. (2022) generates static interaction gestures through rule-based mapping of language commands to specific tasks. Juravsky et al. (2022) utilized BERT (Devlin et al., 2018) to infer language commands, but their method requires pre-defined tasks and different low-level policies for task execution. Wang et al. (2022b) unified various tasks in a CVAE (Yao et al., 2022) network with a language interface, but their performance was limited due to challenges in grounding target objects and contact areas for the characters. Recently, there have been some explorations on LLM-based agent control. Brohan et al. (2023) uses fine-tuned VLM (Vision Language Model) to directly output actions for low-level robots. Rocamonde et al. (2023) employs CLIP-generated cos-similarity as RL training rewards. In contrast, UniHSI utilizes large language models to transfer language commands into the formation of *Chain of Contacts* and design a robust unified controller to execute versatile interaction based on the structured formation.

## 3 METHODOLOGY

As shown in Fig. 2, UniHSI supports versatile human-scene interaction control following language commands. In the following subsections, we first illustrate how we design the unified interaction formulation as CoC(Sec. 3.1). Then we show how we translate language commands into the unified formulation by the LLM Planner (Sec. 3.2). Finally, we elaborate on the construction of the Unified Controller (Sec. 3.3).

### 3.1 CHAIN OF CONTACTS

The initial effort of UniHSI lies in the unified formulation of interaction. Inspired by Hassan et al. (2021b), which infers contact regions of humans and objects based on the interaction gestures of humans, we propose a high correlation between contact regions and interaction types. Further, interactions are not limited to a single gesture but involve sequential transitions. To this end, we can universally define interaction as CoC $\mathcal{C}$, with the formulation as

$$\mathcal{C} = \{\mathcal{S}_1, \mathcal{S}_2, ...\}, \tag{1}$$

where $\mathcal{S}_i$ is the $i^{th}$ contact step. Each step $\mathcal{S}$ includes several contact pairs. For each contact pair, we control whether a joint contacts the corresponding object part and the direction of the contact.

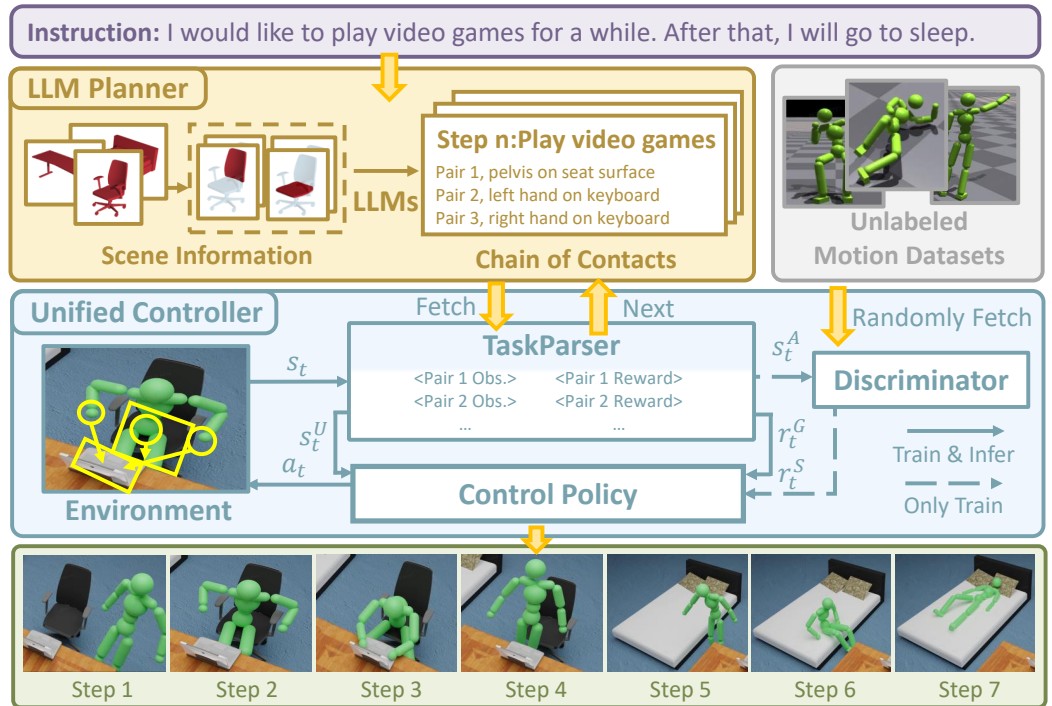

Figure 2: **Comprehensive Overview of UniHSI.** The entire pipeline comprises two principal components: the LLM Planner and the Unified Controller. The LLM Planner processes language inputs and background scenario information to generate multi-step plans in the form of CoC. Subsequently, the Unified Controller executes CoC step by step, producing interaction movements.

We construct each contact pair with five elements: an object $o$, an object part $p$, a humanoid joint $j$, the contact type $c$ of $j$ and $p$, and the relative direction $d$ from $j$ to $p$. The contact type includes "contact", "not contact", and "not care". The relative direction includes "up", "down", "front", "back", "left", and "right". For example, one contact unit $\{o, p, j, c, d\}$ could be {chair, seat surface, pelvis, contact, up}. In this way, we can formulate each $\mathcal{S}$ as

$$\mathcal{S} = \{\{o_1, p_1, j_1, c_1, d_1\}, \{o_2, p_2, j_2, c_2, d_2\}, ...\}. \tag{2}$$

CoC is the output of the LLM Planner and the input of the Unified Controller.

## 3.2 LARGE LANGUAGE MODEL PLANNER

We leverage LLMs as our planners to infer language commands $\mathcal{L}$ into manageable plans $\mathcal{C}$. As shown in Fig. 3, the inputs of the LLM Planner include language commands $\mathcal{L}$, background scenario information $\mathcal{B}$, humanoid joint information $\mathcal{J}$ together with pre-set instructions, rules and examples. Specifically, $\mathcal{B}$ includes several objects $\mathcal{O}$ and their optional spatial layouts. Each object consists of several parts $\mathcal{P}$, *i.e.*, a chair could consist of arms, the back, and the seat. The humanoid joint information is pre-defined for all scenarios. We use prompt engineering to combine these elements together and instruct LLMs to output task plans. By modifying instructions in the prompts, we can generate specified numbers of plans for diverse ways of interactions. We can also let LLMs automatically generate plausible plans given the scenes. In this way, we build our interaction datasets to train and evaluate the Unified Controller.

## 3.3 UNIFIED CONTROLLER

The Unified Controller takes multi-step plans $\mathcal{C}$ and background scenarios in the form of meshes and point clouds as input and outputs realistic movements coherent to the environments.

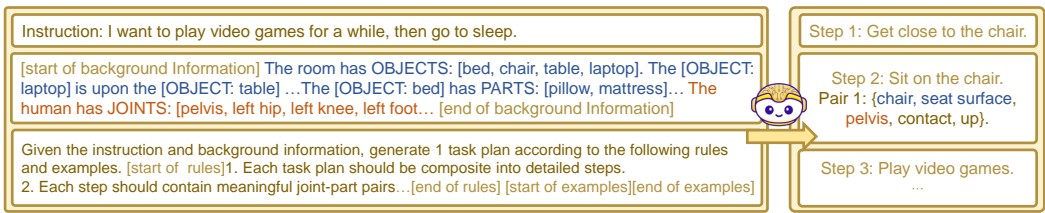

Figure 3: The Procedure for Translating Language Commands into Chains of Contacts.

**Preliminary.** We build the controller upon AMP (Peng et al., 2021). AMP is a goal-conditioned reinforcement learning framework incorporated with an adversarial discriminator to model the motion prior. Its objective is defined by a reward function $R(\cdot)$ as

$$R(\boldsymbol{s}_t, \boldsymbol{a}_t, \boldsymbol{s}_{t+1}, \mathcal{G}) = w^G R^G(\boldsymbol{s}_t, \boldsymbol{a}_t, \boldsymbol{s}_{t+1}, \mathcal{G}) + w^S R^S(\boldsymbol{s}_t, \boldsymbol{s}_{t+1}). \tag{3}$$

The task reward $R^G$ defines the high-level goal $\mathcal{G}$ an agent should achieve. The style reward $R^S$ encourages the agent to imitate low-level behaviors from motion datasets. $w^G$ and $w^S$ are empirical weights of $R^G$ and $R^S$, respectively. $\boldsymbol{s}_t$, $\boldsymbol{a}_t$, $\boldsymbol{s}_{t+1}$ are the state at time $t$, the action at time $t$, the state at time $t+1$, respectively. The style reward $R^S$ is modeled using an adversarial discriminator $D$, which is trained according to the objective:

$$\arg\min_D \; -\mathbb{E}_{d^{\mathcal{M}}(\boldsymbol{s}_t, \boldsymbol{s}_{t+1})} \left[ \log \left( D(\boldsymbol{s}_t^A, \boldsymbol{s}_{t+1}^A) \right) \right] - \mathbb{E}_{d^{\pi}(\boldsymbol{s}, \boldsymbol{s}_{t+1})} \left[ \log \left( 1 - D(\boldsymbol{s}^A, \boldsymbol{s}_{t+1}^A) \right) \right]$$
$$+ w^{\mathrm{gp}} \, \mathbb{E}_{d^{\mathcal{M}}(\boldsymbol{s}, \boldsymbol{s}_{t+1})} \left[ \left\| \left\| \nabla_{\phi} D(\phi) \right|_{\phi = (\boldsymbol{s}^A, \boldsymbol{s}_{t+1}^A)} \right\|^2 \right], \tag{4}$$

where $d^{\mathcal{M}}(\boldsymbol{s}, \boldsymbol{s}_{t+1})$ and $d^{\pi}(\boldsymbol{s}, \boldsymbol{s}_{t+1})$ denote the likelihood of a state transition from $\boldsymbol{s}_t$ to $\boldsymbol{s}_{t+1}$ in the dataset $\mathcal{M}$ and the policy $\pi$ respectively. $w^{\mathrm{gp}}$ is an empirical coefficient to regularize gradient penalty. $\boldsymbol{s}^A = \Phi(\boldsymbol{s})$ is the observation for discriminator. The style reward $r^S = R^S(\cdot)$ for the policy is then formulated as:

$$R^S(\boldsymbol{s}_t, \boldsymbol{s}_{t+1}) = -\log(1 - D(\boldsymbol{s}_t^A, \boldsymbol{s}_{t+1}^A)). \tag{5}$$

We adopt the key design of motion discriminator for realistic motion modeling. In our implementation, we feed 10 adjacent frames together into the discriminator to assess the style. Our main contribution to the controller parts lies in unifying different tasks. As shown in the left part of Fig. 4 (a), AMP (Peng et al., 2021), as well as most of the previous methods (Juravsky et al., 2022; Zhao et al., 2023), design specified task observations, task objectives, and hyperparameters to train task-specified control policy. In contrast, we unify different tasks into Chains of Contacts and devise a TaskParser to process the uniform representation.

**TaskParser.** As the core of the Unified Controller, the TaskParser is responsible for formulating CoC into uniform task observations and task objectives. It also sequentially fetches steps for multi-round interaction execution.

Given one specific contacting pair $\{o, p, j, c, d\}$, for task observation, the TaskParser collects the corresponding position $\boldsymbol{v}^j \in \mathbb{R}^3$ of the joint $j$, and point clouds $\boldsymbol{v}^p \in \mathbb{R}^{m \times 3}$ of the object part $p$ from the simulation environment, where $m$ is the point number of point clouds. It selects the nearest point $\boldsymbol{v}^{np} \in \boldsymbol{v}^p$ from $\boldsymbol{v}^p$ to $\boldsymbol{v}^j$ as the target point for contact. We formulate task observation of the single pair as $\{\boldsymbol{v}^{np} - \boldsymbol{v}^j, c, d\}$. For the task observation in the network, we map $c$ and $d$ into digital numbers, but we still use the same notation for simplicity. Combining these contact pairs together, we get the uniform task observations $s^U = \{\{\boldsymbol{v}_1^{np} - \boldsymbol{v}_1^j, c_1, d_1\}, \{\boldsymbol{v}_2^{np} - \boldsymbol{v}_2^j, c_2, d_2\}, ..., \{\boldsymbol{v}_n^{np} - \boldsymbol{v}_n^j, c_n, d_n\}\}$.

The task reward $r^G = R^G(\cdot)$ is the summarization of all contact pair rewards:

$$R^G = \sum_k w_k R_k, \; k = 1, 2, ..., n. \tag{6}$$

We model each contact reward $R_k$ according to the contact type $c_k$. When $c_k = \mathrm{contact}$, the contact reward encourages the joint $j$ to be close to the part $p$, satisfying the specified direction $d$. When

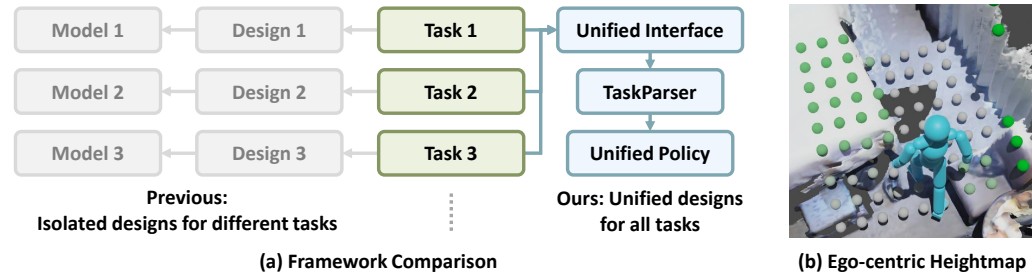

(a) Framework Comparison        (b) Ego-centric Heightmap

Figure 4: **Design Visualization.** (a) Our framework ensures a unified design across tasks using the unified interface and the TaskParser. (b) The ego-centric height map in a ScanNet scene is depicted by green dots, with darker shades indicating greater height.

Table 2: Performance Evaluation on the ScenePlan Dataset.

| Source | Success Rate (%) ↑ | | | Contact Error ↓ | | | Success Steps | | |
|---|---|---|---|---|---|---|---|---|---|
| | Simple | Mid | Hard | Simple | Mid | Hard | Simple | Mid | Hard |
| PartNet (Mo et al., 2019) | 91.1 | 63.2 | 39.7 | 0.038 | 0.073 | 0.101 | 2.3 | 4.5 | 6.1 |
| wo Adaptive Weights | 21.2 | 5.3 | 0.1 | 0.181 | 0.312 | 0.487 | 0.7 | 1.2 | 0.0 |
| wo Heightmap | 61.6 | 45.7 | 0.0 | 0.068 | 0.076 | - | 1.8 | 3.4 | 0.0 |
| ScanNet (Dai et al., 2017) | 76.1 | 43.5 | 32.2 | 0.067 | 0.101 | 0.311 | 1.8 | 2.9 | 4.9 |

$c_k$ = notcontact, we hope the joint $j$ is not close to the part $p$. If $c_k$ = not care, we directly set the reward to max. Following the idea, the $k^{th}$ contact reward $R_k$ is defined as

$$R_k = \begin{cases} w_{\text{dis}}\exp(-w_{dk}\|\boldsymbol{d}_k\|) + w_{\text{dir}}\max(\overline{\boldsymbol{d}}_k\hat{\boldsymbol{d}}_k, 0), & c_k = \text{contact} \\ 1 - \exp(-w_{dk}\|\boldsymbol{d}_k\|), & c_k = \text{not contact} \\ 1, & c_k = \text{not care} \end{cases} \tag{7}$$

where $\boldsymbol{d}_k = \boldsymbol{v}^{np} - \boldsymbol{v}^j$ indicates the $k^{\text{th}}$ distance vector, $\overline{\boldsymbol{d}}_k$ is the normalized unit vector of $\boldsymbol{d}_k$, $\hat{\boldsymbol{d}}_k$ is the unit direction vector specified by direction $d_k$, and $c_k$ is the $k^{\text{th}}$ contact type. $w_{dis}, w_{dir}, w_{dk}$ are corresponding weights. We set the scale interval of $R_k$ as $[0, 1]$ and use *exp* to ensure it.

Similar to the formulation of contact reward, the TaskParser considers a step to be completed if All $k = 1, 2, ..., n$ satisfy: if $c_k$ = contact : $\|\boldsymbol{d}_k\| < 0.1$ and $\overline{\boldsymbol{d}}_k\hat{\boldsymbol{d}}_k > 0.8$, if $c_k$ = not contact : $\|\boldsymbol{d}_k\| > 0.1$, if $c_k$ = not care, $True$.

**Adaptive Contact Weights.** The formulation of 6 includes lots of weights to balance different contact parts of the rewards. Empirically setting them requires much laboring and is not generalizable to versatile tasks. To this end, we adaptively set these weights based on the current optimization process. The basic idea is to give parts of rewards that are hard to optimize high rewards while lowering the weights of easier parts. Given $R_1, R_2, ..., R_n$, we heuristically set their weights to

$$w_k = \frac{1 - R_k}{n - \sum_{k=1,2,...,n} R_k + e}, \tag{8}$$

**Ego-centric Heightmap.** The humanoid must be scene-aware to avoid collision when navigating or interacting in a scene. We adopt similar approaches in Wang et al. (2022a); Won et al. (2022); Starke et al. (2019) that sample surrounding information as the humanoid's observation. We build a square ego-centric heightmap that samples the height of surrounding objects (Fig. 4 (b)). It is important to extend our methods into real scanned scenarios such as ScanNet (Dai et al., 2017) in which various objects are densely distributed and easily collide.

## 4 EXPERIMENTS

Existing methods and datasets related to human-scene interactions mainly focus on short and limited tasks (Hassan et al., 2021a; Peng et al., 2021; Hassan et al., 2023; Wang et al., 2022b). To the best of our knowledge, we are the first method that supports arbitrary horizon interactions with language commands as input. To this end, we construct a novel dataset for training and evaluation. We also conduct various ablations with vanilla baselines and key components of our framework.

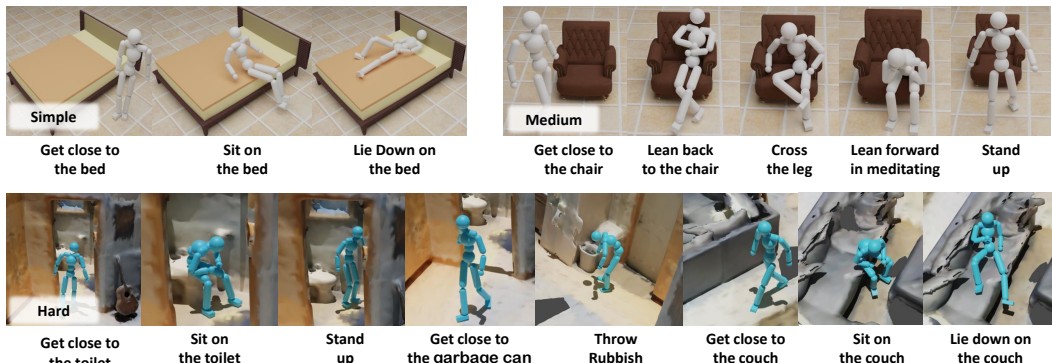

Figure 5: Visual Examples Illustrating Tasks of Varying Difficulty Levels.

## 4.1 DATASETS AND METRICS

To facilitate the training and evaluation of UniHSI, we construct a novel ScenePlan dataset comprising various indoor scenarios and interaction plans. The indoor scenarios are collected and constructed from object datasets and scanned scene datasets. We leverage our LLM Planner to generate interaction plans based on these scenarios. The training of our model also requires motion datasets to train the motion discriminator, which constrains our agents to interact in natural ways. We follow the practice of Hassan et al. (2023) to evaluate the performance of our method.

**ScenePlan.** We gather scenarios for ScenePlan from PartNet (Mo et al., 2019) and ScanNet (Dai et al., 2017) datasets. PartNet offers indoor objects with fine-grained part annotations, ideal for LLM Planners. We select diverse objects from PartNet and compose them into scenarios. For ScanNet, which contains real indoor room scenes, we collect scenes and annotate key object parts based on fragmented area annotations. We then employ the LLM Planner to generate various interaction plans from these scenarios. Our training set includes 40 objects from PartNet, with 5-20 plausible interaction steps generated for each object. During training, we randomly choose 1-4 objects from this set for each scenario and select their steps as interaction plans. The evaluation set consists of 40 PartNet objects and 10 ScanNet scenarios. We construct objects from PartNet into scenarios either manually or randomly. We generated 1,040 interaction plans for PartNet scenarios and 100 interaction plans for ScanNet scenarios. These plans encompass diverse interactions, including different types, horizons, and multiple objects.

**Motion Datasets.** We use the SAMP dataset (Hassan et al., 2021a) and CIRCLE (Araújo et al., 2023) as our motion dataset. SAMP includes 100 minutes of MoCap clips, covering common walking, sitting, and lying down behaviors. CIRCLE contains diverse right and left-hand reaching data. We use all clips in SAMP and pick 20 representative clips in CIRCLE for training.

**Metrics.** We follow Hassan et al. (2023) that uses *Success Rate* and *Contact Error* (*Precision* in Hassan et al. (2023)) as the main metrics to measure the quality of interactions quantitatively. Success Rate records the percentage of trials that humanoids successfully complete every step of the whole plan. In our experiments, we consider a trial of $n$ steps to be successfully completed if humanoids finish it in $n \times 10$ seconds. We also record the average error of all contact pairs:

$$\text{ContactError} = \sum_{i, c_i \neq 0} er_i / \sum_{i, c_i \neq 0} 1, \qquad er_i = \begin{cases} ||\boldsymbol{d}_k||, & c_i = \text{contact} \\ \min(0.3 - ||\boldsymbol{d}_k||, 0). & c_i = \text{not contact} \end{cases} \tag{9}$$

We further record *Success Steps*, which denotes the average success step in task execution.

## 4.2 PERFORMANCE ON SCENEPLAN

We initially conducted experiments on our ScenePlan dataset. To measure performance in detail, we categorize task plans into three levels: simple, medium, and hard. We classify plans within 3 steps as simple tasks, those with more than 3 steps but with a single object as medium-level tasks, and those with multiple objects as hard tasks. Simple task plans typically involve straightforward interactions. Medium-level plans encompass more diverse interactions with multiple rounds of transitions. Hard

Table 3: Ablation Study on Baseline Models and Vanilla Implementations.

| Methods | Success Rate (%) ↑ | | | Contact Error ↓ | | |
|---|---|---|---|---|---|---|
| | Sit | Lie Down | Reach | Sit | Lie Down | Reach |
| NSM - Sit (Starke et al., 2019) | 75.0 | - | - | 0.19 | - | - |
| SAMP - Sit (Hassan et al., 2021a) | 75.0 | - | - | 0.06 | - | - |
| SAMP - Lie Down(Hassan et al., 2021a) | - | 50.0 | - | - | 0.05 | - |
| InterPhys - Sit (Hassan et al., 2023) | 93.7 | - | - | 0.09 | - | - |
| InterPhys - Lie Down(Hassan et al., 2023) | - | 80.0 | - | - | 0.30 | - |
| AMP (Peng et al., 2021)-Sit | 77.3 | - | - | 0.090 | - | - |
| AMP-Lie Down | - | 21.3 | - | - | 0.112 | - |
| AMP-Reach | - | - | **98.1** | - | - | 0.016 |
| AMP-Vanilla Combination (VC) | 62.5 | 20.1 | 90.3 | 0.093 | 0.108 | 0.032 |
| UniHSI | **94.3** | **81.5** | 97.5 | **0.032** | **0.061** | **0.016** |

(a) Visual comparisons on task performance

(b) Comparisons on Success Rate v.s. Training Steps

Figure 6: **Visual Ablations.** (a) Our model exhibits superior natural and accurate performance compared to baselines in tasks such as "Sit" and "Lie Down". (b) Our model demonstrates more efficient and effective training procedures.

task plans introduce multiple objects, requiring agents to navigate between these objects and interact with one or more objects simultaneously. Examples of tasks are illustrated in Fig. 5.

As shown in Table 2, UniHSI performs well in simple task plans, exhibiting a high Success Rate and low Error. However, as task plans become more diverse and complex, the performance of our model experiences a noticeable decline. Nevertheless, the Success Steps metric continues to increase, indicating that our model still performs well in parts of the plans. It's important to note that the scenarios in the ScenePlan test set are unseen during training, and scenes from ScanNet exhibit a modality gap with the training set. The overall performance on the test set demonstrates the versatile capability, robustness, and generalization ability of UniHSI.

## 4.3 ABLATION STUDIES

### 4.3.1 KEY COMPONENTS ABLATION

**Choice of LLMs for UniHSI.** We evaluated different Language Model (LM) choices for the LLM Planner using 100 sets of language commands. We compared task plan Execution Success Rate (ESR) and Planning Correctness (PC) among humans, GPT-3.5OpenAI (2020), and GPT-4OpenAI (2023) across 10 tests per plan. PC is evaluated by humans, with choices of "correct" and "not correct". GPT-4 outperformed GPT-3.5, but both LLMs still lag behind human

Table 4: UniHSI with different LLMs.

| LLM Type | ESR (%) ↑ | PC (%) ↑ |
|---|---|---|
| Human | 73.2 | - |
| w. GPT-3.5 | 35.6 | 49.1 |
| w. GPT-4 | 57.3 | 71.9 |

performance. Failures typically involved incomplete planning and out-of-distribution interactions, like GPT-3.5 occasionally skipping transitions or generating out-of-distribution actions like opening a laptop. While using more rules in prompts and GPT-4 can mitigate these issues, errors can still occur.

**Adaptive Weights.** Table 2 demonstrates that removing Adaptive Weights from our controller leads to a substantial performance decline across all task levels. Adaptive Weights are crucial for optimizing various contact pairs effectively. They automatically adjust weights, reducing them for unused

or easily learned pairs and increasing them for more challenging pairs. This becomes especially vital as tasks become more complex.

**Ego-centric Heightmap.** Removing the Ego-centric Heightmap results in performance degradation, especially for difficult tasks. This heightmap is essential for agent navigation within scenes, enabling perception of surroundings and preventing collisions with objects. This is particularly critical for challenging tasks involving complex scenarios and numerous objects. Additionally, the Ego-centric Heightmap is key to our model's ability to generalize to real scanned scenes.

### 4.3.2 DESIGN COMPARISON WITH PREVIOUS METHODS

**Baseline Settings.** We compared our approach to previous methods using simple interaction tasks like "Sit," "Lie Down," and "Reach." Direct comparisons are challenging due to differences in training data and code unavailability for a closely related method (Hassan et al., 2023). We integrated key design elements from Hassan et al. (2023) into our baseline model (Peng et al., 2021) to ensure fairness. Task observations and objectives were manually formulated for various tasks, following Hassan et al. (2023), with task objectives expressed as:

$$R^G = \begin{cases} 0.7R^{\text{near}} + 0.3R^{\text{far}}, & \text{if distance} > 0.5\text{m} \\ 0.7R^{\text{near}} + 0.3, & \text{otherwise} \end{cases} \tag{10}$$

In this equation, $R^{\text{far}}$ encourages character movement toward the object, and $R^{\text{near}}$ encourages specific task performance when the character is close, necessitating task-specific designs.

We also created a vanilla baseline by consolidating multiple tasks within a single model. We combined task observations from various tasks and included task choices within these observations. We randomly selected tasks and trained them with their respective rewards during training. This experiment involved a total of 70 objects (30 for sitting, 30 for lying down, and 10 for reaching) with 4096 trials per task and random variations in orientation and object placement during evaluation.

**Quantitative Comparison.** In Table 3, UniHSI consistently outperforms or matches baseline implementations across various metrics. The performance advantage is most pronounced in complex tasks, especially the challenging "Lie Down" task. This improvement stems from our approach of breaking tasks into multi-step plans, reducing task complexity. Additionally, our model benefits from shared motion transitions among tasks, enhancing its adaptability. Figure 6 (b) shows that our methods achieve higher success rates and converge faster than baseline implementations. Importantly, the vanilla combination of AMP (Peng et al., 2021) results in a noticeable performance drop in all tasks while our methods remain effective. This difference is because the vanilla combination introduces interference and inefficiencies in training, whereas our approach unifies tasks into consistent representations and objectives, enhancing multi-task learning.

**Qualitative Comparison.** In Figure 6 (a), we qualitatively visualize the performance of baseline methods and our model. Our model performs more naturally and accurately than the baselines in tasks like "Sit" and "Lie Down". This is primarily attributed to the differences in task objectives. Baseline objectives (Eq. 10) model the combination of sub-tasks, such as walking close and sitting down, as simultaneous processes. Consequently, agents tend to perform these different goals simultaneously. For example, they may attempt to sit down even if they are not in the correct position or throw themselves like a projectile onto the bed, disregarding the natural task progression. On the other hand, our methods decompose tasks into natural movements through language planners, resulting in more realistic interactions.

## 5 CONCLUSION

UniHSI is a unified Human-Scene Interaction (HSI) system adept at diverse interactions and language commands. Defined as Chains of Contacts (CoC), interactions involve sequences of human joint-object part contact pairs. UniHSI integrates a Large Language Planner for command translation into CoC and a Unified Controller for uniform execution. Comprehensive experiments showcase UniHSI's effectiveness and generalizability, representing a significant advancement in versatile and user-friendly HSI systems. **Acknowledgement.** We acknowledge Shanghai AI Lab and NTU S-Lab for their funding support.

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
