

| Get close to the chair 1 | Sit on the chair 1 | Rest right hand on the vase | Stand up |

| Get close to the chair 2 | Sit on the chair 2 | Reach the drawer | Stand up |

Figure 1: Illustration of a Multi-Object Interaction Scenario.

## A    LIMITATIONS AND FUTURE WORK.

Apart from the advantages of our framework, there are a few limitations. First, our framework can only control humanoids to interact with fixed objects. We do not take moving or carrying objects into consideration. Enabling humanoids to interact with movable objects is an important future direction. Besides, we do not integrate LLM seamlessly into the training process. In the current design, we use pre-generated plans. Involving LLM in the training pipeline will promote the scalability of interaction types and make the whole framework more integrated.

## B    IMPLEMENTATION DETAILS

We follow Peng et al. (2021) to construct the low-level controller, including a policy and discriminator networks. The policy network comprises a critic network and an actor network, both of which are modeled as a CNN layer followed by two MLP layers with [1024, 1024, 512] units. The discriminator is modeled with two MLP layers having [1024, 1024, 512] units. We use PPO (Schulman et al., 2017) as the base reinforcement learning algorithm for policy training and employ the Adam optimizer Kingma & Ba (2014) with a learning rate of 2e-5. Our experiments are conducted on the IsaacGym (Makoviychuk et al., 2021) simulator using a single Nvidia A100 GPU with 8192 parallel environments.

## C    DETAILED PROMPTING EXAMPLE OF THE LLM PLANNER

As shown in Table. 3. We present the full prompting example of the input and output of the LLM Planner that is demonstrated in Fig. 2 and Fig. 3 of the main paper. The output is generated by OpenAI (2020). Notably, in Tab. 3, example 1 step 2 pair 2: the OBJECT is the chair and PART is the left knee. It's a design choice. Our framework supports interactions between joints. We model the interaction between joints in the same way as the interaction with objects. We only need to replace the point cloud of the object part with a joint position. Some parts of the plans involve "walking to a specific place," which do not contain contacts. To model these special cases in our representations and execute them uniformly, we treat them as a pseudo contact: contacting the pelvis (root) to the target place point. This allows the policy to output a "walking" movement. We represent such cases as {object, none, none, none, direction}. In the future study, we will collect a list of language commands and integrate ChatGPT OpenAI (2020) and GPT OpenAI (2023) into the loop to evaluate the performance of the whole framework of UniHSI.

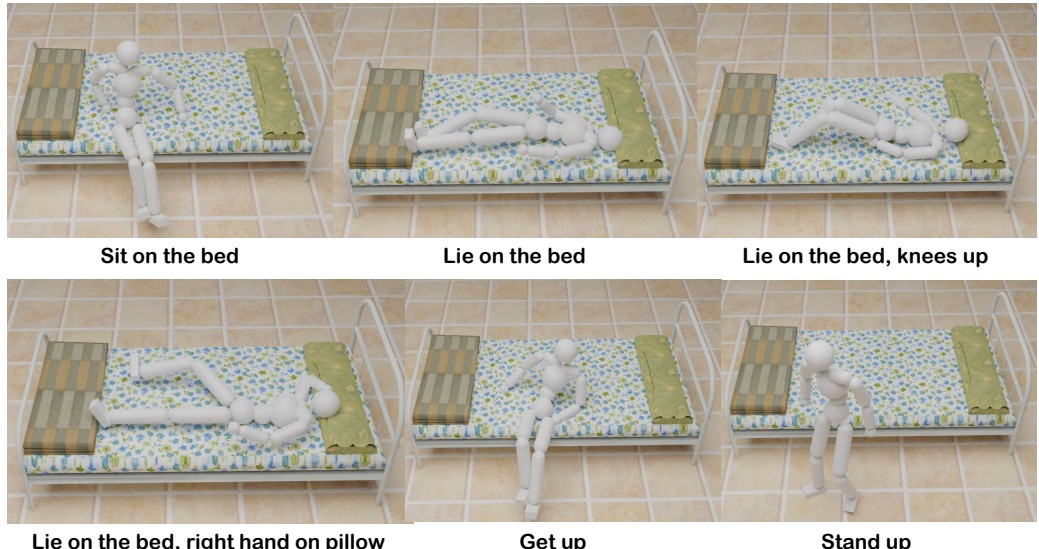

| **Sit on the bed** | **Lie on the bed** | **Lie on the bed, knees up** |

| **Lie on the bed, right hand on pillow** | **Get up** | **Stand up** |

Figure 2: Illustration of a Multi-Step Interaction Involving the Same Object.

## D    DETAILS OF THE SCENEPLAN

We present three examples of different levels of interaction plans in the ScenePlan in Table. 4, 5, and 6, respectively. Simple-level interaction plans involve interactions within 3 steps and with 1 object. Medium-level interaction plans involve more than 3 steps with 1 object. Hard-level interaction plans involve interactions of more than 3 steps and more than 1 object. Specifically, each interaction plan has an item number and two subitems named "obj" and "chain_of_contacts". The "obj" item includes information about objects like object ID, name, and transformation parameters. The "chain_of_contacts" item includes steps of contact pairs in the form of CoC.

We provide the list of interaction types that are included in the training and evaluation of our framework in Table. 7 and 8.

## E    MORE VISUALIZATIONS

We further provide more quantitative results in Fig. 1, 2, 3.

## F    DEMONSTRATION OF FAILURE PLANNING

In Table F, we showcase instances where LLMs encounter challenges in generating accurate plans. We bolded the failure in the plans. Plans produced by LLMs may occasionally falter in spatial relations. This issue is mainly attributed to their suboptimal grasp of spatial contexts. Furthermore, LLMs may occasionally devise plans involving object manipulation, presenting difficulties in successful execution at the current stage.

## G    USER STUDY ON MOTION REALITY.

To examine the global reality of the generated motion, we further conducted a user study on the evaluation of motion reality. The results are presented in the Tab. G. The Naturalness score, ranging from 0 to 5, reflects the degree of perceived naturalness, with higher scores indicating a more natural movements. Similarly, the Semantic Faithfulness score ranges from 0 to 5. A higher score denotes a greater alignment with the semantic input.

However, quantitative evaluation is challenging at this stage and requires further exploration.

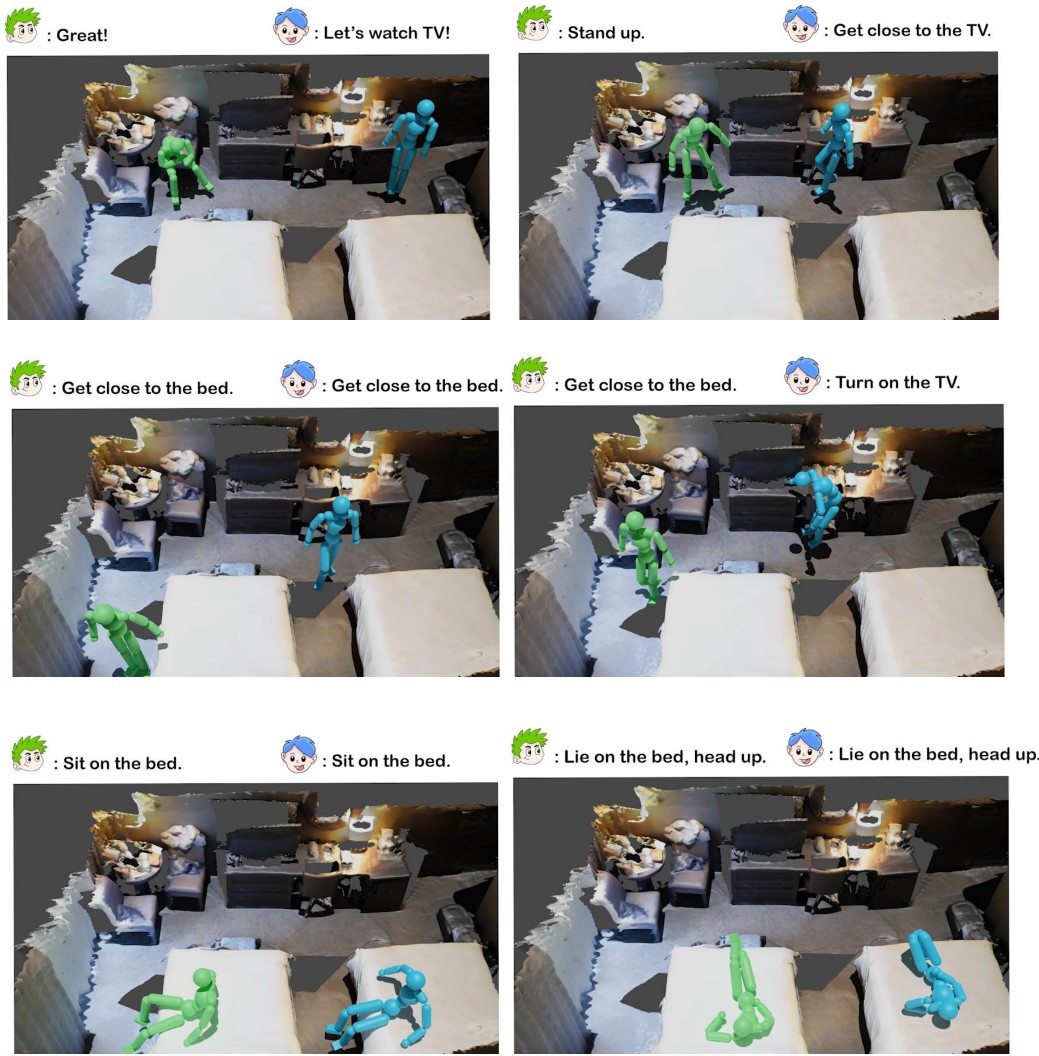

Figure 3: **Illustration of Multi-Agent Interaction.** Note: Achieving "multi-agent interaction" is presently limited to the commands level.

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

Table 1: Demonstration of Failure Planning.

Instruction: I want to get up from the bed and watch movies.

Output

| Human | GPT-3.5 (OpenAI, 2020) | GPT-4 (OpenAI, 2023) |
|---|---|---|
| Step 1: Get up from the bed. Pair 1: {bed, mattress, torso, not contact, up} Pair 2: {bed, mattress, pelvis, contact, up} Step 2: Stand up from the bed. Pair 1: {bed, none, none, none, front} Step 3: Get close to the chair. Pair 1: {chair, none, none, none, front} Step 4: Sit on the chair. Pair 1: {chair, seat surface, pelvis, contact, up} Step 5: control the laptop. Pair 1: {laptop, keyboard, right hand, contact, up} Pair 2: {laptop, keyboard, left hand, contact, up} | Step 1: Get up from the bed. Pair 1: {bed, none, none, none, **up**} Step 2: Get close to the chair. Pair 1: {chair, none, none, none, front} Step 3: **Open the laptop.** Pair 1: {laptop, screen, left hand, contact, up} Pair 2: {laptop, screen, right hand, contact, up} Step 4: Search for movies on the laptop. Pair 1: {laptop, keyboard, left hand, contact, up} Pair 2: {laptop, keyboard, right hand, contact, up} | Step 1: Get up from the bed. Pair 1: {bed, mattress, torso, not contact, up} Step 2: Move away from the bed. Pair 1: {bed, none, none, none, **back**} Step 3: Get close to the chair. Pair 1: {chair, none, none, none, front} Step 4: Sit on the chair. Pair 1: {chair, seat surface, pelvis, contact, up} Step 5: Use the keyboard of the laptop to start a movie. Pair 1: {laptop, keyboard, right hand, contact, up} Pair 2: {laptop, keyboard, left hand, contact, up} |

Table 2: User Study on Motion Reality.

| | Naturalness | Semantic Faithfulness |
|---|---|---|
| AMPPeng et al., 2021-baseline | 3.3 | - |
| UniHSI-PartNetMo et al., 2019 | 4.2 | 4.2 |
| UniHSI-ScanNetDai et al., 2017 | 3.9 | 4.1 |

OpenAI. Gpt-3: Generative pre-trained transformer 3. https://openai.com/research/gpt-3, 2020.

OpenAI. Gpt-4 technical report, 2023.

Xue Bin Peng, Ze Ma, Pieter Abbeel, Sergey Levine, and Angjoo Kanazawa. Amp: Adversarial motion priors for stylized physics-based character control. *ACM Transactions on Graphics (ToG)*, 40(4):1–20, 2021.

John Schulman, Filip Wolski, Prafulla Dhariwal, Alec Radford, and Oleg Klimov. Proximal policy optimization algorithms. *arXiv preprint arXiv:1707.06347*, 2017.

Table 3: **Exemplification of the LLM Planner through Detailed Prompting.** This caption provides a comprehensive illustration of the input and output of the LLM Planner.

| Input |
| --- |

Instruction: I want to play video games for a while, then go to sleep.
Background Information:
[start of background Information]
The room has OBJECTS: [bed, chair, table, laptop].
The [OBJECT: laptop] is upon the [OBJECT: table]. The [OBJECT: table] is in front of the [OBJECT: chair]. The [OBJECT: bed] is several meters away from [OBJECT: table]. The human is several meters away from these objects. The [OBJECT: bed] has PARTS: [pillow, mattress]. The [OBJECT: chair] has PARTS: [back_soft_surface, seat_surface, left_armrest_hard_surface, right_armrest_hard_surface]. The [OBJECT: table] has PARTS: [board]. The [OBJECT: laptop] has PARTS: [screen, keyboard]. The human has JOINTS: [pelvis, left hip, left knee, left foot, right hip, right knee, right foot, torso, head, left shoulder, left elbow, left hand, right shoulder, right elbow, right hand].
[end of background Information]
Given the instruction and background information, generate 1 task plan according to the following rules and examples.
[start of rules]
1. Each task plan should be composite into detailed steps. If the human is not close to the target object, the first step should be to get close to the object.
2. Each step should contain meaningful joint-part pairs.
3. Each joint-part pair should be formatted into {OBJECT, PART, JOINT, Contact type, Contact Direcion}. Or if the step is getting close to an object, the step should be formatted into {none, none, none, none, relative direction of the target object}. JOINT should replace JOINT in the format in the background information. Important: PART in the format should only be replaced by PART or JOINT in the background information. The choices of Contact type include [contact, not contact]. The choices of Contact Direction include [front, back, left, right, up, down, none].
4. Be plausible. Do not generate uncommon interactions.
5. Only interact with still objects. Do not move objects.
[end of rules]
[start of an example]
Instruction: Generate a 6-step interaction with the object.
[start of background Information]
The room has a [OBJECT: chair].
The [OBJECT: chair] has PARTS: [back_soft_surface, armrest_hard_surface, armrest_hard_surface, seat_surface].
The human has JOINTS: [pelvis, left hip, left knee, left foot, right hip, right knee, right foot, torso, head, left shoulder, left elbow, left hand, right shoulder, right elbow, right hand].
The human is several meters away from the [OBJECT: chair].
[end of background Information]
Step 1: Get close to the chair.
Pair 1: {chair, none, none, none, front}
Step 2: Sit on the chair.
Pair 1: {chair, seat_surface, pelvis, contact, up}
Step 3: Rest the right hand on the right arm of the chair.
Pair 1: {chair, seat_surfac, pelvis, contact, up}
Pair 2: {chair, armrest_hard_surface, right hand, contact, up}
Step 4: Place the right foot on the left knee.
Pair 1: {chair, seat_surface, pelvis, contact, up}
Pair 2: {chair, left knee, right foot, contact, up}
Step 5: Stand up.
Pair 1: {chair, none, none, none, front}
Step 6: Sit on the chair.
Pair 1: {chair, seat_surface, pelvis, contact, up}
[end of an example]

| Output |
| --- |

Step 1: Get close to the chair.
Pair 1: {chair, none, none, none, front}
Step 2: Sit on the chair.
Pair 1: {chair, seat_surface, pelvis, contact, up}
Step 3: Use the laptop to play video games.
Pair 1: {chair, seat_surface, pelvis, contact, up}
Pair 2: {laptop, keyboard, left hand, contact, up}
Pair 3: {laptop, keyboard, right hand, contact, up}
Step 4: Stand up from the chair.
Pair 1: {chair, none, none, none, front}
Step 5: Get close to the bed.
Pair 1: {bed, none, none, none, front}
Step 6: Sit on the bed.
Pair 1: {bed, mattress, pelvis, contact, down}
Step 7: Lie down on the bed.
Pair 1: {bed, mattress, pelvis, contact, down}
Pair 2: {bed, mattress, torso, contact, up}
Pair 3: {bed, pillow, head, contact, up}

Table 4: **Illustration of Simple-Level Interaction Plans in ScenePlan.** Simple-level interaction plans encompass interactions within three steps and involve a single object.

```
{
    "0000":
    {
        "obj":
        {
            "000":
            {
                "id": "12747",
                "name": "bed",
                "rotate": [[1.5707963267948966, 0, 0], [0, 0, -1.5707963267948966]],
                "scale": 2.5,
                "transfer": [0,-2,0],
            }
        },
        "chain_of_contacts": [[["bed000", "none", "none", "none", "front"]],
                             [["bed000", "mattress25", "pelvis", "contact", "up"],
                                    ["bed000", "mattress25", "head", "not contact", "up"]],
                             [["bed000", "mattress25", "pelvis", "contact", "up"],
                                    ["bed000", "mattress25", "left_foot", "contact", "up"],
                                    ["bed000", "mattress25", "right_foot", "contact", "up"],
                                    ["bed000", "mattress25", "head", "contact", "up"]]]
    }
}
```

Table 5: **Exemplar of Medium-Level Interaction Plans in ScenePlan.** Medium-level interaction plans encompass interactions exceeding three steps and involving a single object.

```
{
    "0000":
    {
        "obj": {
            "000":{
                "id": "45005",
                "name": "chair",
                "rotate": [[1.5707963267948966, 0, 0], [0, 0, -1.5707963267948966]],
                "scale": 1.5,
                "transfer": [0,-2,0],
            }
        },
        "chain_of_contacts": [[["chair000", "none", "none", "none", "front"]],
                             [["chair000", "seat_soft_surface42", "pelvis", "contact", "up"]],
                             [["chair000", "seat_soft_surface42", "pelvis", "contact", "up"],
                             ["chair000", "back_soft_surface47", "torso", "contact", "none"]],
                             [["chair000", "seat_soft_surface42", "pelvis", "contact", "up"],
                             ["chair000", "back_soft_surface47", "torso", "contact", "none"]],
                             [["chair000", "seat_soft_surface42", "pelvis", "contact", "up"],
                             ["chair000", "arm_sofa_style44", "left_hand", "contact", "up"],
                             ["chair000", "arm_sofa_style48", "right_hand", "contact", "up"]],
                             [["chair000", "seat_soft_surface42", "pelvis", "contact", "up"],
                             ["chair000", "arm_sofa_style44", "left_hand", "not contact", "up"],
                             ["chair000", "arm_sofa_style48", "right_hand", "not contact", "up"]],
                             [["chair000", "seat_soft_surface42", "pelvis", "contact", "up"],
                             ["chair000", "left_knee", "right_foot", "contact", "none"]],
                             [["chair000", "seat_soft_surface42", "pelvis", "contact", "up"],
                             ["chair000", "back_soft_surface47", "torso", "not contact", "none"]],
                             [["chair000", "none", "none", "none", "front"]]]}
    }
}
```

Table 6: **An example of hard-level interaction plans in ScenePlan.** Hard-level interaction plans involve interactions of more than 3 steps and more than 1 object.

```
{
    "0000":
    {
        "obj":
        {
            "000":
            {
                "id": "37825",
                "name": "chair",
                "rotate": [[1.5707963267948966, 0, 0], [0, 0, -1.5707963267948966]],
                "scale": 1.5,
                "transfer": [0,-2,0]
            },
            "001":
            {
                "id": "21980",
                "name": "table",
                "rotate": [[1.5707963267948966, 0, 0], [0, 0, 1.5707963267948966]],
                "scale": 1.8,
                "transfer": [1,-2,0]
            },
            "002":
            {
                "id": "11873",
                "name": "laptop",
                "rotate": [[1.5707963267948966, 0, 0], [0, 0, 1.5707963267948966]],
                "scale": 0.6,
                "transfer": [0.8,-2,0.65]
            },
            "003":
            {
                "id": "10873",
                "name": "bed",
                "rotate": [[1.5707963267948966, 0, 0], [0, 0, -1.5707963267948966]],
                "scale": 3,
                "transfer": [-0.2,-4,0]
            }
        },
        "chain_of_contacts": [[["chair000", "none", "none", "none", "front"]],
                            [["chair000", "seat_soft_surface58", "pelvis", "contact", "up"]],
                            [["chair000", "seat_soft_surface58", "pelvis", "contact", "up"],
                                ["laptop002", "keyboard15", "left_hand", "contact", "none"],
                                ["laptop002", "keyboard15", "right_hand", "contact", "none"]],
                            [["chair000", "none", "none", "none", "front"]],
                            [["bed003", "none", "none", "none", "front"]],
                            [["bed003", "mattress16", "pelvis", "contact", "up"],
                                ["bed003", "mattress16", "head", "not contact", "up"]],
                            [["bed003", "mattress16", "pelvis", "contact", "up"],
                                ["bed003", "mattress16", "left_foot", "contact", "up"],
                                ["bed003", "mattress16", "right_foot", "contact", "up"],
                                ["bed003", "pillow17", "head", "contact", "up"]],
                            [["bed003", "mattress16", "pelvis", "contact", "up"],
                                ["bed003", "mattress16", "head", "not contact", "up"]],
                                [["bed003", "none", "none", "none", "front"]]]
    }
}
```

Table 7: List of Interactions in ScenePlan-1

| Interaction Type | Contact Formation |
|---|---|
| Get close to xxx | {xxx, none, none, none, dir} |
| Stand up | {xxx, none, none, none, dir} |
| Left hand reaches xxx | {xxx, part, left_hand, contact, dir} |
| Right hand reaches xxx | {xxx, part, right_hand, contact, dir} |
| Both hands reaches xxx | {{xxx, part, left_hand, contact, dir}, {xxx, part, right_hand, contact, dir}} |
| Sit on xxx | {xxx, seat_surface, pelvis, contact, up} |
| Sit on xxx, left hand on left arm | {{xxx, seat_surface, pelvis, contact, up}, {xxx, left_arm, left_hand, contact, up}} |
| Sit on xxx, right hand on right arm | {{xxx, seat_surface, pelvis, contact, up}, {xxx, right_arm, right_hand, contact, up}} |
| Sit on xxx, hands on arms | {{xxx, seat_surface, pelvis, contact, up}, {xxx, left_arm, left_hand, contact, none}, {xxx, right_arm, right_hand, contact, none}} |
| Sit on xxx, hands away from arms | {{xxx, seat_surface, pelvis, contact, up}, {xxx, left_arm, left_hand, not contact, none}, {xxx, right_arm, right_hand, not contact, none}} |
| Sit on xxx, left elbow on left arm | {{xxx, seat_surface, pelvis, contact, up}, {xxx, left_arm, left_elbow, contact, up}} |
| Sit on xxx, right elbow on right arm | {{xxx, seat_surface, pelvis, contact, up}, {xxx, right_arm, right_elbow, contact, up}} |
| Sit on xxx, elbows on arms | {{xxx, seat_surface, pelvis, contact, up}, {xxx, left_arm, left_elbow, contact, none}, {xxx, right_arm, right_elbow, contact, none}} |
| Sit on xxx, left hand on left knee | {{xxx, seat_surface, pelvis, contact, up}, {xxx, left_knee, left_hand, contact, up}} |
| Sit on xxx, right hand on right knee | {{xxx, seat_surface, pelvis, contact, up}, {xxx, right_knee, right_hand, contact, up}} |
| Sit on xxx, hands on knees | {{xxx, seat_surface, pelvis, contact, up}, {xxx, left_knee, left_hand, contact, none}, {xxx, right_knee, right_hand, contact, none}} |
| Sit on xxx, left hand on stomach | {{xxx, seat_surface, pelvis, contact, up}, {xxx, pelvis, left_hand, contact, none}} |
| Sit on xxx, right hand on stomach | {{xxx, seat_surface, pelvis, contact, up}, {xxx, pelvis, right_hand, contact, none}} |
| Sit on xxx, hands on stomach | {{xxx, seat_surface, pelvis, contact, up}, {xxx, pelvis, left_hand, contact, none}, {xxx, pelvis, right_hand, contact, none}} |
| Sit on xxx, left foot on right knee | {{xxx, seat_surface, pelvis, contact, up}, {xxx, right_knee, left_foot, contact, none}} |
| Sit on xxx, right foot on left knee | {{xxx, seat_surface, pelvis, contact, up}, {xxx, left_knee, right_foot, contact, none}} |
| Sit on xxx, lean forward | {{xxx, seat_surface, pelvis, contact, up}, {xxx, back_surface, torso, not contact, none}} |
| Sit on xxx, lean backward | {{xxx, seat_surface, pelvis, contact, up}, {xxx, back_surface, torso, contact, none}} |

Table 8: List of Interactions in ScenePlan-2

| Interaction Type | Contact Formation |
| --- | --- |
| Lie on xxx | {{xxx, mattress, pelvis, contact, up}, {xxx, pillow, head, contact, up}} |
| Lie on xxx, left knee up | {{xxx, mattress, pelvis, contact, up}, {xxx, pillow, head, contact, up {xxx, mattress, left_knee, not contact, none}} |
| Lie on xxx, right knee up | {{xxx, mattress, pelvis, contact, up}, {xxx, pillow, head, contact, up}, {xxx, mattress, right_knee, not contact, none}} |
| Lie on xxx, knees up | {{xxx, mattress, pelvis, contact, up}, {xxx, pillow, head, contact, up}, {xxx, mattress, left_knee, not contact, none}, {xxx, mattress, right_knee, not contact, none}} |
| Lie on xxx, left hand on pillow | {{xxx, mattress, pelvis, contact, up}, {xxx, pillow, head, contact, up}, {xxx, pillow, left_hand, contact, none}} |
| Lie on xxx, right hand on pillow | {{xxx, mattress, pelvis, contact, up}, {xxx, pillow, head, contact, up}, {xxx, pillow, right_hand, contact, none}} |
| Lie on xxx, hands on pillow | {{xxx, mattress, pelvis, contact, up}, {xxx, pillow, head, contact, up}, {xxx, pillow, left_hand, contact, none}, {xxx, pillow, right_hand, contact, none}} |
| Lie on xxx, on left side | {{xxx, mattress, pelvis, contact, up}, {xxx, pillow, head, contact, up}, {xxx, mattress, right_shoulder, not contact, none}} |
| Lie on xxx, on right side | {{xxx, mattress, pelvis, contact, up}, {xxx, pillow, head, contact, up}, {xxx, mattress, left_shoulder, not contact, none}} |
| Lie on xxx, left foot on right knee | {{xxx, mattress, pelvis, contact, up}, {xxx, pillow, head, contact, up}, {xxx, right_knee, left_foot, contact, up}} |
| Lie on xxx, right foot on left knee | {{xxx, mattress, pelvis, contact, up}, {xxx, pillow, head, contact, up}, {xxx, left_knee, right_foot, contact, up}} |
| Lie on xxx, head up | {{xxx, mattress, pelvis, contact, up}, {xxx, pillow, head, not contact, none}} |