# OpenReview forum: "Unified Human-Scene Interaction via Prompted Chain-of-Contacts"
_ICLR.cc/2024/Conference — ICLR 2024 spotlight_

### Official Review · Reviewer_JJQ3 · 2023-10-30

**Soundness:** 3 good
**Presentation:** 3 good
**Contribution:** 4 excellent
**Rating:** 10
**Confidence:** 4

**Summary:**

This paper presents UniHSI, a unified framework for Human-Scene Interaction (HSI). The framework aims to support versatile interaction control through language commands, providing a user-friendly interface. The authors propose a definition of interaction as a Chain of Contacts (CoC), which represents the steps of human joint-object part pairs. UniHSI consists of a Large Language Model (LLM) Planner that translates language prompts into CoC task plans and a Unified Controller that executes the plans. The framework enables diverse interactions, fine-granularity control, and long-horizon transitions. The authors collect a dataset ScenePlan for training and evaluation, and experiments demonstrate the effectiveness and generalizability of the framework.

**Strengths:**

- The proposed method is the first to generate long-horizon and open-domain human-scene interactions.
- The proposed method outperforms existing baselines in both success rate and contact error.
- The method proposed to use an LLM Planner to generate a sequence of contact pairs (Chain of Contacts) as an intermediate representation, and parse that into a reward function for learning a unified controller. I find this solution both intuitive and elegant.

**Weaknesses:**

The paper does not provide a thorough explanation of the method. Specifically, it is unclear how different cases are handled by the LLM Planner, and how each type of interaction pair is translated to the reward. See questions.

**Questions:**

- In supp tbl 1, example 1 step 2 pair 2: The OBJECT is chair and PART is left knee. Is this a typo or a design choice?
- what does {none, none, none, none, front} and {bed, none, none, none, front} mean, and how does the UniHSI controller use this information?
- In supp tbl 1 example 2, why is the direction between mattress and body parts “down” and the direction between pillow and head “up”?
- Can the LLM Planner, or a quick extension of the LLM Planner support fine-grained spatial relationships such as distances and directions between objects? For example, can this method handle cases where 1) a chair is placed too closed to a table so that it is impossible to sit on it,  and 2) two chairs are placed facing opposite directions, and the agent must choose the correct one to watch tv?

---

> ### Author Response · Authors · 2023-11-18
> **Response to Reviewer JJQ3**
>
> Thank you for acknowledging our efforts. Detailed responses are as below.
>
> > Q1: In supp tbl 1, example 1 step 2 pair 2: The OBJECT is chair and PART is left knee. Is this a typo or a design choice?
>
> It is a design choice. Our framework naturally supports interactions between joints. We model the interaction between joints in the same way as interaction with objects. We only need to replace the point cloud of the object part with a joint position. Here the OBJECT is meaningless and can be replaced by "none".
>
> > Q2&Q3: What does {none, none, none, none, front} and {bed, none, none, none, front} mean, and how does the UniHSI controller use this information? Why is the direction between mattress and body parts “down” and the direction between pillow and head “up”?
>
> Some parts of the plans involve "walking to a specific place," which does not contain contacts. To model these special cases in our representations and execute them uniformly, we treat them as a pseudo contact: contacting the pelvis (root) to the target place point. This allows the policy to output a "walking" movement. We represent such cases as {object, none, none, none, direction}. We apologize for some typos in Supp. Tab. 1: {none, none, none, none, front} -> {chair, none, none, none, front}. "down" -> "up". We have revised them.
>
> > Q4: Can the LLM Planner, or a quick extension of the LLM Planner support fine-grained spatial relationships such as distances and directions between objects?
>
> Exploring fine-grained spatial relationships is an intriguing and crucial direction for future study. Our methods currently leverage LLMs like ChatGPT as planners, which possess basic inference abilities on human-scene interaction but may be limited for fine-grained spatial relations. We believe a deeper understanding of representations such as point clouds is necessary. Notably, recent advancements in LLM-driven methods on point cloud understanding (e.g., OpenScene (Peng et al. 2022), PointLLM (Xu et al. 2023)) could be leveraged to extend our methods for more intricate scene reasoning.

---

> > ### Comment · Reviewer_JJQ3 · 2023-11-22
> >
> > Thank you for the clarification. I recommend clarifying the prompt design in better detail in either the main paper or the supplementary material to eliminate potential misunderstandings like mine.

---

> > > ### Author Response · Authors · 2023-11-22
> > > **Response to Reviewer JJQ3**
> > >
> > > Thanks for your suggestions. We have updated the details in the Supp. C.

---

### Official Review · Reviewer_YGEs · 2023-10-30

**Soundness:** 2 fair
**Presentation:** 2 fair
**Contribution:** 3 good
**Rating:** 5
**Confidence:** 2

**Summary:**

The paper proposes to interact with virtual scene interfaces via language commands. Its proposed UniHSI prompts an LLM to translate user commands into (human part, touched object) sequences. An example is "play video games" -> (chair, seat surface, pelvis, contact, down)
-> (table, keyboard, left hand, contact, down) -> (table, keyboard,
right hand, contact, down). They are then executed by a second model.
This modularity allows modeling more interactions than before, and doing to automatically. The authors also propose a new dataset, Sceneplan, which combines PartNet and ScanNet into the above sequences of multiple interactions.

**Strengths:**

* The new Sceneplan dataset could be useful for the community, and the different difficulty levels can help drive the community
* The UniHCI method is the first that generalizes to much more interaction combinations than previous methods (this is claimed by the authors, I am not inside the field enough to judge it) ## Weaknesses
* The evaluation of the UniHSI method would benefit from more ablations on the design choices (loss, which module leads to the error, used motion style datasets) and comparison to more SOTA baselines (here, the authors blame code unavailability and too different training setups)
* The proposed new method (UniHSI) is only evaluated on the proposed new dataset (Sceneplan) which both use the LLM Planner module. This may give an unfair advantage.
* The Sceneplan dataset was automatically generated, although the involved LLM Planner in UniHSI gives much lower performance than humans (57.3 vs 73.2, "Failures typically involved incomplete planning and out-of-distribution interactions"). This may lower the dataset quality severly.
* Most sentences in the paper are hard to parse, mostly because of over-complicated words (see Questions). This can be fixed in a (major) revision.
* The reference section is inconsistent: Six different spellings for CVPR; Harvey 2020 misses where it was published; Zhang 2022a misses the arxiv identifier; in Tevet 2023 ICLR should not be referenced to as only "ICLR, 2023". This can be fixed in a revision.
* The paper may be more applicable to computer vision conferences, judging by the references and by the fact that it is heavily practical/conceptual. This potential weakness can be better judged by the AC and Senior AC.

**Weaknesses:**

See Questions below.

**Questions:**

In Table 3, is there no other SOTA method that allows combinations of Sit, Lie Down, and Reach (other than the AMP vanilla baseline)? Maybe other reviewers from the field could also comment.

Examples of hard-to-parse sentences:
* "The framework is built upon the definition of interaction as Chain of Contacts (CoC): steps of human joint-object part pairs, which is inspired by the strong correlation between interaction types and human-object contact regions."  -> These are three sentences in one.
How about "Interactions intuitively describe contacts between human joints and object parts. Thus, we define sequences of such contacts as Chain of Contacts (CoC)."
* "the initial effort involves the uniform definition of different interactions. We propose that interaction itself contains a strong prior in the form of human-object contact regions." -> It is confusing to use "uniform" in a non-statistical way when the surrounding text contains priors etc.. What do you mean by uniform throughout the text?
* "To facilitate the unified execution, we devise the TaskParser as the core of the Unified Controller." -> Please use simpler language.
* Section 3.2 is easier to read. It would be great to write more like in this section.

In how far is the discriminator $D$ adversarial and what effect does this have?

In Equation 7, instead of using exp to ensure positivity, you could consider using Softplus, which is sometimes more numerically stable.

Please analyse the dataset quality in more detail.

Please format the reference section coherently.

Please add the Limitations from the Appendix to the main paper

It would be an option to split the dataset from the method. This way, you could focus more on either of them. E.g., a big multi-round dataset with high quality would be a great contribution.

---

> ### Author Response · Authors · 2023-11-18
> **Response to Reviewer YGEs**
>
> Thanks for your valuable comments. Detailed responses are as below.
>
> > W1: The evaluation of the UniHSI method would benefit from more ablations on the design choices and comparison to more SOTA baselines.
>
> Thanks for your suggestions. We ablated the key designs (i.e. unified architecture) with baseline implementations (see Tab. 3), and technical designs (i.e. Adaptive Weights, Heightmap, LLM choice, see Tab. 2 and Tab. 4). For other settings like loss, motion datasets, we follow the convention of previous methods (AMP [Peng et al. 2022], InterPhys [Hassan et al. 2023]), so we do not further ablate these choices.
>
> > W2: The proposed new method (UniHSI) is only evaluated on the proposed new dataset.
>
> To the best of our knowledge, we are the first method that supports arbitrary horizon interactions with language commands as input. There are few methods or datasets for exact comparison. We propose the new dataset for evaluation purposes. We will release the dataset later to support further comparison. Additionally, we compare our method with previous approaches on traditional tasks (Tab. 3).
>
> > W3: The Sceneplan dataset was automatically generated by LLM, which may lower the dataset quality.
>
> We have done post-processing to filter out failure plans (i.e., out-of-distribution plans that move objects) of the ScenePlan dataset. All plans are deemed reasonable for execution. As for the generated plan quality, we have conducted ablations on the effect of choices of LLMs (see Tab. 4).
>
> > W4: Most sentences in the paper are hard to parse, mostly because of over-complicated words (see Questions). This can be fixed in a (major) revision.
>
> Thank you for your feedback. We have revised the paper and will keep polishing the language.
>
> > W5: The reference section is inconsistent: Six different spellings for CVPR; Harvey 2020 misses where it was published; Zhang 2022a misses the arxiv identifier; in Tevet 2023 ICLR should not be referenced to as only "ICLR, 2023". This can be fixed in a revision.
>
> We appreciate your attention to detail. The references have been revised to follow a consistent format sourced from Google Scholar.
>
> > W6: The paper may be more applicable to computer vision conferences.
>
> We believe ICLR is an inclusive conference welcoming submissions from all areas of machine learning. Our topic is applicable to the "applications to robotics, autonomy, planning" track.
>
> > Q1: In Table 3, is there no other SOTA method that allows combinations of Sit, Lie Down, and Reach (other than the AMP vanilla baseline)
>
> To our knowledge, InterPhys [Hassan et al. 2023] is the most recently published work in this field, but without open-source code.
>
> > Q2: "The initial effort involves the uniform ...". What do you mean by uniform throughout the text?
>
> Here "uniform" means different interactions can be defined in the same way.
>
> > Q3:  "To facilitate the unified execution, we devise the TaskParser as the core of the Unified Controller." -> Please use simpler language.
>
> We revised it to "We design the TaskParser to support the unified execution. It serves as the core of the Unified Controller."
>
> > Q4: In how far is the discriminator D adversarial and what effect does this have?
>
> The discriminator plays a crucial role. Its adversarial nature contributes to learning the style from the motion datasets.
>
> > Q5: In Equation 7, instead of using exp to ensure positivity, you could consider using Softplus, which is sometimes more numerically stable.
>
> As our goal is not only to ensure positivity but also to constrain the output value within the range of [0,1], we believe using exp remains a better choice in this context.
>
> > Q6: Please analyze the dataset quality in more detail.
>
> As responded in W3, we apply post-processing to filter out failure cases to ensure the quality of plans in the ScenePlan dataset. All plans are deemed reasonable for execution.
>
> > Q7: Please add the Limitations from the Appendix to the main paper.
>
> We will merge the limitations from the Appendix into the main paper, and provide a more comprehensive discussion in the final version.
>
> > Q8: It would be an option to split the dataset from the method. This way, you could focus more on either of them. E.g., a big multi-round dataset with high quality would be a great contribution.
>
> Thank you for the suggestion. The initial motivation was to develop the dataset for a comprehensive evaluation of the new method. We will explore a more rigorous and high-quality multi-round dataset in future work.

---

### Official Review · Reviewer_tgmd · 2023-10-30

**Soundness:** 3 good
**Presentation:** 3 good
**Contribution:** 3 good
**Rating:** 8
**Confidence:** 4

**Summary:**

The paper proposes a method to generate physical human-scene interactions from language inputs. The method is capable of generating multiple tasks with a single model.

A large language model (LLM) is used to translate the language prompt (e.g go sit on the chair) into a plan of multiple sequential steps (e.g. go closer to the chair, make contact between hip and seating area in the chair). Each step is defined as the contact between a human joint and an object part. The paper calls this: chain of contact.

The authors derive observation and reward based on the contact pair. A humanoid agent is trained using AMP (Peng et al. 2021) to maximize the sum of rewards for all the steps specified by the LLM.

To train the humanoid agent, the authors collected a dataset of plans by prompting the LLM in different scenarios. These scenarios require access 3D scene data (i.e. object meshes, scene layout, object parts).

**Strengths:**

The paper is novel, tackles an important problem, provides interesting results, opens the door for exciting future directions, is easy to read, and is technically sound.
- The main novelties of the paper are:

1. Using LLM as a task planner. i.e. using LLM to convert textual prompts into smaller executable steps.
2. Representing all the steps as contact pairs. i.e chain of contact
3. Unifying the design of the observation and reward for all tasks.

- The method is capable of generating multiple interactions with the same model.

**Weaknesses:**

- The motion quality looks worse than previous works like InterPhys, AMP, or NSM.
- The paper mentions the use of prompt engineering to generate the plan data. It is not entirely clear how was this done. How difficult was this? How sensitive is the model to changes in the prompt? I think this process involves several steps which were not discussed in detail. Thus I think it will be hard to reproduce this part.
- The LLM expects examples of a task plan with each prompt. It is unclear how many examples were needed in total. How were these examples generated? manually?
- The comparison in Table 3 seems weak. I understand the difficulty since the code for InterPhys is unavailable. However, I still have many questions about the comparison. How many test objects were used and how many trails were run? Did you use similar numbers as the ones used by InterPhys Hassan et al 2023? In addition, the performance of AMP seems quite low. InterPhys is an extension of AMP so I expect AMP to perform better than the provided numbers. I suspect the model has not been trained properly.
- The method does not handle interaction with moving objects.

Minor weaknesses\comments for improvements
- There are three recent(possibly concurrent) works that use LLM as an agent planner. It would be helpful to the reader to discuss the differences between the proposed methods and their approaches.
1. Athanasiou et al."SINC: Spatial Composition of 3D Human Motions for Simultaneous Action Generation"
2. Brohan et al."RT-2: New model translates vision and language into action"
3. Rocamonde† et al. "VISION-LANGUAGE MODELS ARE ZERO-SHOT REWARD MODELS FOR REINFORCEMENT LEARNING"
- The authors argue for the value of language input. Language is useful for some applications but it is not necessarily a better or more user-friendly interface. For many users, a joystick, brush, and fingers might be more comfortable. Especially in games or digital design.
- The contact pair contains 5 elements. Then it is not a pair. Maybe it is better to be called a contact tuple or a set.
- It would be nice to see an example of when the human plan succeeded but the LLM one. It would be useful to put both plans side by side

**Questions:**

- Can this approach be applied to kinematic models? What would need to change?
- What is the importance of having a reward for the "not care" contact?
- What is the difference between $\hat{d_k}$ and $\bar{d_k}$? I don't see how the directional part of the reward in Eq.7 enforces the desired behavior.
- In Table 2. How are the weights chosen without adaptive weights?

---

> ### Author Response · Authors · 2023-11-18
> **Response to Reviewer tgmd (1/2)**
>
> Thank you for acknowledging our efforts. Detailed responses are as below.
>
> > W1: The motion quality looks worse than previous works like InterPhys, AMP, or NSM.
>
> This work focuses on unifying diverse interactions into a single model. We did not apply specific optimizations for sub-tasks. So the motion quality might be slightly worse than previous works in some situations. We will keep optimizing this part.
>
> > W2: The paper mentions the use of prompt engineering to generate the plan data. It is not entirely clear how was this done. How sensitive is the model to changes in the prompt?
>
> Please refer to related descriptions in Sec 4.1 "ScenePlan", Supp. Tab. 7-10 for details.
> It can be briefly summarized in 3 steps to generate and process the plan data:
> 1. Use prompts to generate plans in the format shown in Supp. Tab. 7.
> 2. Use scripts to generate data structure from plans in step 1, and output format like Supp. Tab. 8-10.
> 3. Process the data structure by TaskParser and execute the plan, which is described in Sec 3.3.
> Also, we will release the code and models for reproduction.
> Our model in the current stage is sensitive to incomplete or out-of-distribution plans (i.e. plans that skip some key interaction steps or have out-of-distribution interactions). We use prompts to avoid generating these plans as much as possible. However, as shown in Tab.4, we can not completely eliminate such problems.
>
> > W3: It is unclear how many examples were needed in total. How were these examples generated? manually?
>
> The use of an example prompt is for formatting purposes, providing LLMs with the necessary information about the desired output structure. Only one manually-designed example is needed to convey the format to LLMs.
>
> > W4: The comparison in Table 3 seems weak.
>
> Our experiments involved a total of 70 objects (30 for sitting, 30 for lying down, and 10 for reaching) with 4096 trials per task, and random variations in orientation and object placement during evaluation. As InterPhys did not provide evaluation datasets, our settings differ. As for the low performance of the AMP baseline, we keep the same training setting and basic framework except for key components in UniHSI, so we expect the training to be proper. We notice that Interscene [Pan, et al. 2023] also has done similar experiments that extend AMP to sitting tasks. It achieves a Success Rate even worse than we do. We guess it might take a very long time before complex tasks (like "lying down") can converge in the InterPhys way.
>
> > W5:  The method does not handle interaction with moving objects.
>
> We acknowledge this limitation. Interaction with dynamic scenes is a crucial area that we plan to explore in future studies.
>
> > W6: There are three recent(possibly concurrent) works that use LLM as an agent planner. It would be helpful to the reader to discuss the differences between the proposed methods and their approaches.
>
> Thanks for your suggestions. We have added the discussion to our paper.
> 1. Athanasiou et al. "SINC: Spatial Composition of 3D Human Motions for Simultaneous Action Generation":
> It is an interesting kinematics method that synthesizes actions defined by textual descriptions. While SINC synthesizes actions based on textual descriptions, it lacks interaction with objects. Our work focuses on the physical plausibility of interactions with objects.
> 2. Brohan et al. "RT-2: New model translates vision and language into action". Both RT-2 and our method leverage LLMs, but our emphasis differs. RT-2 explores the potential of LMs in a broader context, feeding multimodal inputs into the VLMs. In contrast, we use LLMs as translators for low-level controllers without the need for extensive fine-tuning, offering flexibility in applications.
> 3. Rocamonde† et al. "VISION-LANGUAGE MODELS ARE ZERO-SHOT REWARD MODELS FOR REINFORCEMENT LEARNING". It is also an interesting attempt at the potential of VLMs to guide RL training. This approach employs CLIP-generated cos-similarity as RL training rewards. While similar in using LLMs, our method generates rewards from a unified intermediate representation. We believe both ideas have their potential.
>
> > W7: The authors argue for the value of language input. Language is useful for some applications but it is not necessarily a better or more user-friendly interface.
>
> Agreed. We think it is a matter of high-level control and low-level control. We can use the language as input to direct a movie. We can use a joystick to control game agents. They have irreplaceable value to their applications.
>
> > W8: The contact pair contains 5 elements. Then it is not a pair. Maybe it is better to be called a contact tuple or a set.
>
> Thanks for your suggestions. Since we want to highlight the relation between the "human joint" and "object part", we would argue that "pair" is more suitable.

---

> ### Author Response · Authors · 2023-11-18
> **Response to Reviewer tgmd (2/2)**
>
> > W9: It would be nice to see an example of when the human plan succeeded but the LLM one. It would be useful to put both plans side by side.
>
> Thanks for your suggestions. Please refer to Tab. 5 in the revised paper.
>
> > Q1: Can this approach be applied to kinematic models? What would need to change?
>
> We believe that the high-level concept of leveraging LLMs to infer a unified intermediate representation for downstream tasks is applicable across various fields. Regarding specific kinematic models, there are related works (COINS [Zhao et al. 2022], COUCH [Zhang et al. 2022]) that also employed the idea of contacts to guide interactions but did not go further. The primary challenge in kinematic models lies in maintaining physical plausibility and object coherence. Thus we believe one of the changes is to integrate the recent progress (i.e. [OpenScene, Peng et al. 2022]) in the scene understanding into our framework.
>
> > Q2: What is the importance of having a reward for the "not care" contact?
>
> In situations where we do not want to specify the contact state of certain joints, we set the reward to 1 directly. This ensures that these joints are not considered in the optimization process, especially since adaptive weights are employed.
>
> > Q3: What is the difference between $\hat{d}_k$ and $\overline{d}_k$? I don't see how the directional part of the reward in Eq.7 enforces the desired behavior.
>
> $\hat{d}_k$ represents the desired direction, while $\overline{d}_k$ is the real direction (normalized). By calculating the Dot Product of these two vectors, we enforce the real direction to approach the desired direction, ensuring alignment and desired behavior. However, there is a typo that "min" should be "max".
>
> > Q4: In Table 2. How are the weights chosen without adaptive weights?
>
> We simply adopt average weights.

---

### Official Review · Reviewer_gSqi · 2023-11-01

**Soundness:** 2 fair
**Presentation:** 4 excellent
**Contribution:** 3 good
**Rating:** 6
**Confidence:** 3

**Summary:**

The paper contributes the first unified framework that can generate physically plausible HSI motions in static scenes given compounded language commands. To examine the method's effectiveness, the paper constructs a new dataset containing thousands of interaction plans on existing 3D scenes. Experiments show that the proposed framework outperforms baselines on different single-step tasks, and also exhibits its ability to handle hard task plans and good generalizability.

**Strengths:**

(1) The proposed framework is unified without task-specific designs. It can directly support language instruction input without manual task decompositions.

(2) The method achieves impressive performance on unseen real-world scenarios, indicating its good generalizability.

(3) The paper is well-organized and easy to follow.

**Weaknesses:**

(1) The evaluation metrics aim to examine whether a method can complete the task. However, both the Success Rate and the Contact Error are defined on $c_k,d_k$ that are generated by the LLM Planner. If the LLM Planner fails to generate reasonable chains of contacts or contact pairs, the evaluation may be misleading.

(2) The current evaluation focuses on contact information that only relates to local human body parts. I think it is also important to examine the global reality of the generated motion. The evaluation for motion reality may include whether the motion is human-like and whether it is semantically faithful to the language command.

(3) In the definition of task observations, the human joint $v_k^j$ corresponds to the nearest object surface point $v_k^{np}$. However, $v_k^{np}$ may not be a reasonable contact point for $v_k^j$. Understanding object geometries and affordances is important to finding ideal contact areas that human prefers.

(4) The method is limited to static scenes.

**Questions:**

* In Figure 5, there are some interactions among different human joints such as "cross the leg" and "lean forward in meditating". How to model such interactions in the method?

* In the method design, the style reward examines the reality of every two adjacent frames. I am curious whether it is sufficient without evaluating longer frame ranges that include more integral human motion semantics.

---

> ### Author Response · Authors · 2023-11-18
> **Response to Reviewer gSqi**
>
> Thanks for your valuable comments. Detailed responses are as below.
>
> > W1: If the LLM Planner fails to generate reasonable chains of contacts or contact pairs, the evaluation may be misleading.
>
> The evaluation metrics primarily assess the versatile execution ability of the unified controller under reasonable plans. To evaluate the performance with LLMs in the loop, we include the success rate of plan executions in Table 4. Additionally, we conducted a user study to evaluate the Planning Faithfulness metric in Table 4 to measure the correctness of the LLM Planner. A quantitative joint assessment of the whole framework is challenging at this stage and requires further exploration.
>
> > W2: The current evaluation focuses on contact information that only relates to local human body parts.
>
> Thank you for your insights. To examine the global reality of the generated motion, we further conducted a user study on the evaluation of motion reality. The results are presented in the Supp. Tab. 6. The Naturalness score, ranging from 0 to 5, reflects the degree of perceived naturalness, with higher scores indicating a more natural appearance. Similarly, the Semantic Faithfulness score ranges from 0 to 5. A higher score denotes a greater alignment with the semantic input. However, quantitative evaluation is challenging at this stage and requires further exploration.
>
> > W3: In the definition of task observations, the human joint corresponds to the nearest object surface point. However, it may not be a reasonable contact point. Understanding object geometries and affordances is important to finding ideal contact areas that human prefers.
>
> The nearest object surface point of the contact part is enough for basic interactions. However, we acknowledge that it is limited when it comes to further refined interactions like holding a cup. Understanding object geometries and affordances is crucial, and we plan to explore this in future studies to enhance the method's capability.
>
> > W4: The method is limited to static scenes.
>
> Agreed. Interaction with dynamic scenes is an important direction for exploration, and we are studying to accommodate dynamic environments in the following project.
>
> > Q1: There are some interactions among different human joints such as "cross the leg" and "lean forward in meditating". How to model such interactions in the method.
>
> Our framework naturally supports interactions between joints. We model interactions between joints in the same manner as interactions with objects. To achieve this, we replace the point cloud of the object part with the position of the joint.
>
> > Q2: In the method design, the style reward examines the reality of every two adjacent frames. I am curious whether it is sufficient without evaluating longer frame ranges that include more integral human motion semantics.
>
> Thank you for your insights. In our implementation, we feed 10 adjacent frames together into the discriminator to assess the style. We have revised the paper for clarification.

---

### Author Response · Authors · 2023-11-18
**Common Response**

Dear ICLR Reviewers,

&nbsp;

We would like to express our sincere gratitude for your time and effort in reviewing our paper. Your constructive feedback has been invaluable in enhancing the quality of our work.

We have carefully addressed each of your comments and provided detailed responses in our revised manuscript. Additionally, we have made updates to the paper to incorporate your suggestions, and these revisions are highlighted in $\color{red}{red}$ for your convenience.

We eagerly anticipate your continued feedback and hope that you find the updated manuscript to be satisfactory.

Thank you once again for your commitment.

&nbsp;

Best regards,

Authors of 2155

---

### Meta-Review · Area_Chair_AwwB · 2023-12-07

**Metareview:**

The paper presents a unified framework for Human-Scene Interaction, which employs an LLM planner to unfold a natural language command as a chain of contact (CoC) and CoC is then executed by a controller. The reviewers think the work is novel, the dataset resulted from the work is useful, and the empirical results are impressive. Reviewer tgmd stated that "The paper is novel, tackles an important problem, provides interesting results, opens the door for exciting future directions, is easy to read, and is technically sound." Reviewer gSqi mentioned "The method achieves impressive performance on unseen real-world scenarios, indicating its good generalizability." Reviewer JJQ3 mentioned "The proposed method is the first to generate long-horizon and open-domain human-scene interactions." Reviewer YGEs, who gave the lowest rating among the reviewers, also has made many points about the strength of the paper, e.g., "The new Sceneplan dataset could be useful for the community, and the different difficulty levels can help drive the community". Reviewer YGEs is concerned that that the work might be more suitable for a CV venue -- the AC thinks the work should be of interest to the ICLR audience. That said, the paper is not without weaknesses. The reviewers brought up an extensive list of issues with technical and presentation details. The authors did a good job at addressing them in the responses and in the revision.

**Justification For Why Not Higher Score:**

Conceptually, the paper follows a typical approach of LLM-planning followed by a domain-specific executor.

**Justification For Why Not Lower Score:**

The paper is novel in addressing the specific domain and produces impressive results.

---

### Decision · Program_Chairs · 2024-01-16

Accept (spotlight)